# Quality Control of Pesticide Residue Measurements and Evaluation of Their Results

**DOI:** 10.3390/molecules28030954

**Published:** 2023-01-18

**Authors:** Árpád Ambrus, Vy Vy Ngoc Doan, Júlia Szenczi-Cseh, Henriett Szemánné-Dobrik, Adrienn Vásárhelyi

**Affiliations:** 1Doctoral School of Nutrition and Food Sciences, University of Debrecen, 4032 Debrecen, Hungary; 2Southern Pesticide Control and Testing Center, Plant Protection Department, 71007 Ho Chi Minh City, Vietnam; 3Freelancer Food Safety Adviser, 1116 Budapest, Hungary; 4Food Chain Safety Centre, Non-profit Ltd., Pesticide Residue Analytical Laboratory, 3529 Miskolc, Hungary; 5Food Chain Safety Laboratory Directorate, 1095 Budapest, Hungary

**Keywords:** pesticide residues, quality control procedures, sources of errors of residue analyses, reproducibility of results

## Abstract

Pesticide residues are monitored in many countries around the world. The main aims of the programs are to provide data for dietary exposure assessment of consumers to pesticide residues and for verifying the compliance of the residue concentrations in food with the national or international maximum residue limits. Accurate residue data are required to reach valid conclusions in both cases. The validity of the analytical results can be achieved by the implementation of suitable quality control protocols during sampling and determination of pesticide residues. To enable the evaluation of the reliability of the results, it is not sufficient to test and report the recovery, linearity of calibration, the limit of detection/quantification, and MS detection conditions. The analysts should also pay attention to and possibly report the selection of the portion of sample material extracted and the residue components according to the purpose of the work, quality of calibration, accuracy of standard solutions, and reproducibility of the entire laboratory phase of the determination of pesticide residues. The sources of errors potentially affecting the measured residue values and the methods for controlling them are considered in this article.

## 1. Introduction

A sufficient amount of safe food cannot be provided for the continuously growing population of the world without the use of pesticides at the current technological level. The global demand for, and the production as well as the use of pesticides have increased steadily during the past decades and are projected to continue growing [1,2]. Pesticides are chemical substances with various degrees of toxicity and modes of action [3,4]. To control the target pests certain concentrations of pesticide residues must remain in/on the treated species. Consumers are generally concerned about the toxic chemicals in their food. According to the survey conducted by the European Food Safety Authority, pesticide residues in food (40%) and antibiotic, hormone, or steroid residues in meat (39%) are the main food safety-related concerns among Europeans [5].

To protect consumers and the environment, the national authorities authorize the use of pesticides only after the critical evaluation of their toxicity, biological efficacy and residues remaining in/on food as well as in the environment [6,7,8,9,10,11]. The OECD Guidelines for Testing of Chemicals are a collection of the most relevant internationally agreed testing methods used by government, industry, and independent laboratories [12]. They are intended to enhance the validity and international acceptance of test data and reduce unnecessarily repeated tests [13,14,15]. Many non-OECD member countries adopt the same principles [12] or give permission for use only after [16,17,18] a pesticide active ingredient has been authorized by countries having an advanced registration system [8,10,11]. To facilitate international trade and assist the national registration authorities to establish their own limits, the CODEX maximum residue limits, MRLs, are elaborated by the FAO/WHO Joint Meeting on Pesticide Residues, JMPR, [19], further considered in a stepwise procedure by the Codex Committee on Pesticide Residues, and approved by the Codex Alimentarius Commission [20,21].

To control the safe and efficient use of pesticides, their residues are regularly monitored in food and environmental samples in many countries according to risk-based sampling plans [22,23,24,25,26,27,28,29,30] or targeted surveillance with limited scope and sampling targets. For example, the world-wide activities are demonstrated with some selected publications from Argentina to Vietnam [31,32,33,34,35,36,37,38,39,40,41,42,43,44,45,46]. In the European Union the largest number of residues tested within the EU-coordinated and national pesticide residue monitoring programs were reported in 2020 by Luxembourg (659), Malta (643), Germany (626), France (619) and Belgium (617) [47]. Concerning all 30 countries reporting their monitoring results to EFSA, multiple residues were detected in 27.2% of the samples, and 30%, 22.3%, 4.1%, 0.5%, and 0.02% of samples contained 0, 1, 5, 10 and 15 different residues, respectively. However, in extreme cases 18 and 31 residues were detected in single strawberry [47] and honeysuckle samples [48]. These results underlined the importance of applying screening methods of the widest possible scope with low limit of detection/limit of quantification (LOD/LOQ) values. For this purpose, good progress has been made in expanding the scope of the methods [49,50,51].

Most publications referenced above [30,31,32,33,34,35,36,37,38,39,40,41,42,43,44,45,46,47,48,49,50] mainly reported minor modifications in sample preparation procedures of the original QuEChERS (Quick, Easy, Cheap, Effective, Rugged, and Safe) method [52] and often provided details of the conditions of the MS mass spectrometry (MS) detection. Other authors reported various combinations of sample preparation [53,54,55,56,57,58,59,60,61]. The authors typically stated the recoveries, linearity, LOD and LOQ values, matrix effects and compared them to the acceptance criteria specified in the major guidance documents [62,63,64]. On the other hand, none of them provided information on the details of sampling, efficiency of subsampling and comminution affecting the reproducibility of the results, or accuracy of reference standard solutions, albeit these steps can be major hidden sources of random and systematic errors [65,66,67,68,69,70].

Drawing realistic conclusions and making appropriate corrective actions can only be done if the monitoring results are accurate and derived from the analyses of samples taken according to the specific objectives of the program. That can only be achieved by implementing rigorous internal quality control of the whole process of the determination of pesticide residues. The basic quality requirements for the monitoring results are defined in five major guidance documents [62,63,64,70,71]. However, several potential hidden errors are not explicitly addressed in these documents. Although over the last two decades several scientific publications have highlighted the effect of these errors on the accuracy and uncertainty of the measurement results [65,67,68,72,73,74,75], the actions for limiting them have rarely been reported in the monitoring studies. Therefore, the reliability of these study results cannot be assessed. Table 1 summarizes the main steps of residue analyses and gives examples for the sources of potential errors.

Every laboratory should introduce and implement appropriate quality control procedures to assure that the results of the analyses are as accurate as possible, and that their uncertainties are kept as low as practical. The random error indicated by the combined relative uncertainty of the results (CV_R_) is influenced by four main factors (Equation (1)): sampling (S), laboratory sample handling including subsampling of large crops (CV_SS_), comminution (CV_Sp_), test portion selection and analyses of sample extracts (CV_A_) [66].
(1)$${\mathrm{CV}}_{R}=\sqrt{{\mathrm{CV}}_{S}^{2}+{\mathrm{CV}}_{\mathrm{SS}}^{2}+{\mathrm{CV}}_{\mathrm{Sp}}^{2}+{\mathrm{CV}}_{A}^{2}}$$

The CV_R_ incorporates the relative precision of all steps of the determination of pesticide residues including sampling.

The analysts usually only report the within-laboratory repeatability/reproducibility of steps from the extraction of the test portions (CV_A_). On the other hand, the reproducibility (CV_L_) is the parameter that realistically characterizes the laboratory measurements including all steps from subsampling to the quantitative determination of residues.
(2)$${\mathrm{CV}}_{L}=\sqrt{{\mathrm{CV}}_{\mathrm{SS}}^{2}+{\mathrm{CV}}_{\mathrm{Sp}}^{2}+{\mathrm{CV}}_{A}^{2}}$$

The analyses phase can be further subdivided into extraction (Ex), clean-up (Cl), evaporation (Ev), and chromatographic determination (Ch):(3)$${\mathrm{CV}}_{A}=\sqrt{{\mathrm{CV}}_{\mathrm{Ex}}^{2}+{\mathrm{CV}}_{\mathrm{Cl}}^{2}+{\mathrm{CV}}_{\mathrm{Ev}}^{2}+{\mathrm{CV}}_{\mathrm{Ch}}^{2}}$$

However, the individual quantification of the contributions of the steps affecting CV_A_ can only be done in practice with applying isotope labelled compounds in specialized laboratory conditions with specific detection instruments [74]. Therefore, their combined effect should be determined in practice with repeated recovery tests (CV_A_) performed at the concentration range that is expected to occur in the samples. Such tests reflect only the effect of operations carried out after spiking the test portions. If the tests are carried out on different days by different analysts the calculated relative standard deviation of the results will only indicate an interim reproducibility of the analyses step, but it is not equivalent to CV_L_ as defined by Equation (2). The results of recovery tests can be used to characterize the within-laboratory reproducibility (CV_L_) only if they are performed with samples containing incurred residues derived from the prior application of a pesticide [65,72,74]. 

Our objectives are to call attention to the hidden errors in the analyses of pesticide residues that can significantly affect the accuracy and reliability of the results. Without aiming for a full review of the vast amount of published data, we describe some practical options for the quality control actions that the program managers can get implemented by the laboratory staff to obtain accurate results with quantified uncertainty. 

## 2. Methods

### 2.1. Sampling

The main objectives of the monitoring program, and in general the analyses of samples, are to obtain correct information with known uncertainty on the pesticide residue levels in the sampling targets and not only in the sample. It is generally recognized that the accuracy and validity of analytical results cannot be better than that of the samples analyzed. The sampling designs and methods are widely described in the scientific literature. Their coverage is beyond the scope of this paper. Briefly, for the monitoring of the pesticide residues in plant commodities and soil, stratified random sampling is the best choice. The sampling target (the area from where the samples are to be collected) can be stratified for instance according to crop, cultivation mode, growing season, soil type, etc. Random samples should be separately taken from each stratum. The minimum number of primary samples to be collected for one composite sample (sample size) depends on the objectives of the program. For instance, the provisions of the Codex sampling standard [70] for the minimum number of primary samples and total mass of a composite laboratory sample should be satisfied where the compliance with MRLs is assessed in goods offered for sale. It is not sufficient to collect [76], for instance, 4 pieces of head cabbages or Chinese cabbages (instead of the minimum five specified in the Codex GL) even though their total mass may be well over 10 kg and 4–5 kg, respectively, which are much larger than the specified minimum of 2 kg. A larger number of primary samples may be collected than the minimum, provided that their representative part can be effectively comminuted with the available laboratory equipment.

The sampling uncertainty is inversely proportional to the sample size (n, the number of primary samples) and depends on the variability of residues in crop units or in single sample increments (CV_1_):(4)$${\mathrm{CV}}_{S}=\frac{{\mathrm{CV}}_{1}}{n}$$

The variability of residues in individual crop units derived from a single field (called within field variability) is close to 80–100-fold [77,78], therefore increasing ‘n’ will decrease the uncertainty of the results and improve the accuracy of the estimated average residue in the sample. Under typical growing conditions the relative uncertainty of sampling is in the range of 25–40% for samples of size 10 and 5, respectively [79,80]. These uncertainties shall be considered when a product is tested before export.

### 2.2. Selection of Portion of Sample to Be Analysed

For testing compliance with MRLs, the portion of commodities specified in the Codex CAC/GL-41-1993 standard should be considered [81]. However, for providing data to estimate the dietary exposure of consumers, the edible portion of commodities should be analyzed. Since the edible portion varies and for instance depends on the variety, maturity of the crop, and local practices for its consumption. Consequently, the specific way of selecting the edible portions should be precisely described in the publications to enable the comparison of the results with other studies.

Most of the operations required for the preparation of the test portions depend very much on the actual condition of the test item and cannot be generally standardized. The laboratory assistants should be well trained on the principles enabling them to perform the tasks properly. Inconsistent operation may lead to high, uncontrollable variability (unquantifiable uncertainty) of the results. For instance, the way of removing adhering soil from root vegetables or outer, withered leaves from leafy vegetables can substantially influence the residues measured. The outer leaves usually contain much higher residues than the inner leaves. Therefore, only the loose leaves should be removed (Figure 1) otherwise the measured residues will not correctly reflect the residue content and may lead to dispute if the lot is repeatedly sampled along the commercial chain.

To prepare samples for the analyses of the edible portion the peeling of fruits with inedible peel should be made in a way that the edible part is not cross contaminated by the residues being on the peel. Large fruits (e.g., watermelon, pumpkin, jackfruit) should be cut into wedge-shaped sections and the flesh part removed with proper spoons as shown in Figure 2. It should be noted that for checking compliance with MRLs, the whole fruit shall be comminuted and further processed. It is recommended that one section from each of the five large crops making up one composite sample according to the Codex sampling standard [70] is used for determining the residues in edible portions and a second set of five sections is comminuted for determination of residues in/on whole fruits.

### 2.3. Subsampling and Comminution of Selected Sample Portions

Because of the usually very large difference in the concentration of residues in individual crop units [78], the whole laboratory sample or representative part of each primary sample (crop unit) must be processed to obtain accurate information on the average residue in the laboratory sample. Omeroglu [82] and Ambrus [83] provided detailed graphical illustrations for obtaining representative subsamples and calculation of CV_L_.

The distribution of residues within the natural crop units is also uneven. For instance, the residues concentrate on the lower part of fruits hanging on the trees or vines due to the runoff of the sprays. Therefore, slices should never be cut from crop units (Figure 3).

Obtaining representative portion of the large crop units (e.g., cabbage, watermelon, papaya, etc.) requires special attention making sure that each crop unit is proportionally represented in the subsample to be comminuted. Figure 4 illustrates the subsampling of large fruits.

The efficiency of cutting, blending of the sample materials may vary from day-to-day and sample-to-sample because of the changing physical properties and textural composition of crops depending on the variety and maturity. Moreover, it is strongly influenced by the sharpness of cutting blades. The fundamental sampling error defined by Gy [84] can be applied for characterizing the relative variance of the residues in comminuted materials [85].
(5)$${\mathrm{CV}}_{\mathrm{Sp}}^{2}=\frac{C\times d^{3}}{w}$$

In Equation (5), the C is the sampling constant depending on the nature of the homogenized material, d is the diameter of the 95th percentile of the comminuted particles, and w is the mass of the test portion. Though CV_Sp_ cannot be calculated for plant materials applying Gy’s theory, Equation (5) clearly indicates the importance of particle size (d^3^) distribution. Reducing the particle size in a comminuted laboratory sample considerably reduces CV_Sp_ and consequently CV_L_ (Equation (2)). Therefore, the proper homogeneity of the comminuted materials should be checked for each sample.

A very quick and convenient method for this purpose is the ‘Petri dish‘ test, in which a small portion of the comminuted material is spread on the glass surface and the particle size distribution is visually checked. If the particles are smaller than 2 mm, the homogeneity would be generally sufficient to keep CV_Sp_ smaller than 10–12% if 10–15 g test portions are taken for extraction [73,86]. Otherwise, the comminutions should be continued preferably by adding a further portion of dry ice [75,87]. Figure 5 provides some examples. Much smaller particles can be obtained, and considerably reduced test portions can be used when liquid nitrogen is used for cryogenic processing [88,89,90]. The two-stage sample processing can also be used in the combination of pre-homogenization of a large sample with proper choppers (C_L_), then transferring its representative 100–150 g portion into a Waring laboratory blender (or a baker if Ultra Turrax is used for fine cutting), and adding about 10% known amount of distilled water for fine comminution (C_F_) [83].

It is generally recommended to add a small portion of water to dry materials to improve the efficiency of comminution [62,63]. The exact amount of added water shall be accounted for in reporting the residue concentrations. The portions for further processing should be taken without delay in small increments (preferably > 10) of the test material from various positions of the blender to obtain representative test portion and avoid segregation.

The CV_Sp_ will be determined by the combined effects of the two comminution steps.
(6)$${\mathrm{CV}}_{\mathrm{Sp}}=\sqrt{\frac{C_{L}}{w_{L}}+\frac{C_{F}}{w_{F}}}$$

Equation (6) should also be applied for estimation of sample processing uncertainty in case of two-stage processing with liquid nitrogen [88,89]. The CV_Sp_ will depend on the C_L_/w_L_ ratio. It is misleading to report the repeatability/reproducibility based on the analyses of spiked portions taken after fine comminution with liquid nitrogen.

The size of test portion significantly affects the reproducibility of the measurements. Based on Gy’s sampling theory the relationship between the mass of the comminuted laboratory sample (m_L_), the test portion (m_TP_), and the CV_Sp_ can be described as [85]:(7)$${\mathrm{CV}}_{\mathrm{Sp}}^{2}={\mathrm{Cd}}^{3}\left(\frac{1}{m_{\mathrm{Tp}}}-\frac{1}{m_{L}}\right)$$

Table 2 shows the change of CV_Sp_ depending on the test portion size taken from the same comminuted material.

Table 2 indicates that reducing the test portion size from 15 g to 1 g will increase the CV_Sp_ by about 3.2 times. For instance, if the CV_Sp_ is 12.2% when a 10 g test portion containing incurred residues is extracted, and then one gramme portions are also taken from the same comminuted matrix, the theoretically expected CV_Sp_ would be about 38.7%. Naturally, the measurable CV_L_ will depend on the combined contribution of CV_Sp_ and CV_A_ according to Equation (2) (CV_SS_ is zero in this case). Provided that the CV_A_ from recovery tests is 10%, and the CV_Sp_-s from Table 2 are 12.2% and 38.7%, the corresponding CV_L_ would be 15.8% and 40%, respectively, if 10 g and 1 g test portions were extracted from the same comminuted material. This significant effect remains unnoticed when the recoveries are determined with spiking the test portions. Therefore, making use of the high sensitivity of the recent MS systems and extracting 1–2 g test portions should only be done after careful checking of the reproducibility of the method with incurred residues, otherwise the real variability of the results may not be reflected [91,92,93]. A practical solution is to extract 5–10 g portions and dilute the extracts to utilize the sensitive detection and reducing the matrix effect [45,49].

### 2.4. Definition of Residues

Where the toxic metabolites or degradation products are present in a treated commodity in toxicologically significant proportion, they should be considered for the determination of the dietary exposure of consumers to pesticide residues. The principles are explained, for instance, in the FAO/WHO JMPR Manual [93]. The analyses of polar metabolites that are often present in conjugated form requires specific procedures and cannot be determined with the usual multi-residue methods. To facilitate testing the compliance with MRLs carried out in large number of samples, the regulatory authorities often establish different definitions of residues for monitoring and risk assessment purposes. The JMPR emphasized that the definition of residues for enforcement purposes should be as practical as possible and preferably based on a single residue component (the parent compound, a metabolite, or a derivative produced in an analytical procedure) as an indicator of the total significant residue, and it should be determinable with a multi-residue procedure whenever possible [93]. Some examples for the different residue definitions are highlighted in Table 3 and Table 4 [94,95].

The definition of residues in commodities of animal origin is often much more complex. The list of Codex MRLs indicates the residues to be tested for checking compliance with MRLs and for risk assessment purposes [21]. The latest recommendations of the JMPR can be found in the JMPR reports [96]. Alternately, the proper composition of residues can be accessed from the websites of the national registration authorities [10,97].

The examples above underline the importance of adhering to the residue definition that fits for the objectives of the study in order to obtain accurate results. Due to the inclusion of metabolites the total residue for risk assessment purposes can be much higher than that for monitoring purposes. In such cases, the calculation of estimated daily intake (EDI) based on the residues defined for monitoring purposes will underestimate the real exposure of consumers and result in wrong conclusions.

### 2.5. Extraction of Residues and Cleanup of Extracts

The selection of solvents and adjusting the pH to obtain acceptable recoveries have been extensively studied, providing sufficient information for the optimization of the procedures for various matrix-analyte combinations. A detailed guidance document for testing the efficiency of extraction [98] provides the basis, if followed, for obtaining accurate results. The efficiency of extraction should always be tested with incurred residues in all kinds of samples.

### 2.6. Accuracy of Standard Solutions

It is evident for every analyst that the accuracy of standard solutions is one of the very basic pre-conditions for the correct quantification of the residues. We cannot assume that the analytical standard prepared in our laboratory is accurate unless it is verified. To assist the laboratories participating in EU proficiency tests to find out the reasons for unsatisfactory results, the EU Reference Laboratory for Pesticide Residues in Fruits and Vegetables organized a ring test for the determination of the concentrations of certified pesticide analytical standards provided in a mixture. Forty official and national reference laboratories from 20 countries took part in the tests [99]. The summary of results is given in Table 5. The accuracy and uncertainty of the analytical standards may be affected by their storage and handling conditions.

All laboratory equipment used for the preparation of analytical standards have their own inherent uncertainty of the nominal volume that is combined with the variability of filling them to mark depending on the daily performance of the analysts. Various manufacturers provide volumetric glassware of different grades. The relative uncertainty of the measured volume can be calculated from their specified tolerance (e.g., 50 ± 0.05 mL) assuming triangular distribution [100],
u = 0.05/√6 = 0.02 mL, CV = 0.02/50 = 4.08 × 10^−4^

The combined uncertainty (CV_exp_) of volumetric measurements can be calculated from the tolerance of the glassware (CV_T_) and the variability of filling them to mark (CV_fil_):(8)$${\mathrm{CV}}_{\exp}=\sqrt{{\mathrm{CV}}_{\mathrm{fil}}^{2}+{\mathrm{CV}}_{T}^{2}}$$

Involving our technicians making most accurately the volumetric measurements based on prior tests, we determined the relative uncertainties of filling in the volumetric glassware [100]. An example of the results is given in Table 6.

Using our five-digit analytical balances, the weighing relative uncertainty of 25 mL water is 1.6 × 10^−6^. It is three magnitudes lower than the volumetric measurement (Table 6). Therefore, the diluted standard solutions should be prepared based on weighing except the last step where an A-grade ≥ 25 mL volumetric flask should be used to obtain the standard concentration in mass/volume (e.g., μg/mL) [65].

We tested the reproducibility of the preparation of diluted standard solutions with the combination of weighing and volumetric measurements according to the regular practice in two of our laboratories [101]. The relative differences were calculated for the nominal concentrations. Some of the results are summarized in Table 7.

The results indicated that the analytical standards can occasionally deviate by 10% from the nominal concentration even with the most careful and precise preparation. The relative uncertainty of nominal concentration (CV_Rep_) increases with the increasing dilution of the standard solutions. These findings underline the importance of verifying the accuracy of analytical standard solutions. As a minimum, two new solutions should be prepared independently and their average chromatographic responses from minimum 5 replicate injections should be compared. If their relative difference is less than 10%, the two solutions can be combined for use. If the difference is larger, then a 3rd solution should be prepared, and the two closest ones can be combined.

The same procedure can be used for comparing the old standard solution with the new one. The SANTE Guidance document suggests accepting the two solutions (old and new or two new ones) if their relative difference is less than 10% [62]. It is pointed out that the *t*-test comparing the mean values cannot be used for this purpose, because it is designed to prove that the two mean values are not significantly different. Instead, the two-sample *t*-test (TOST) should be used to correctly verify that the relative difference between the two mean values is ≤10% [102]. As an alternative to the relatively complicated calculations, Figure 6 can be used for visually testing that the two standard solutions are within the targeted range (Δ_rd_ ≤ 10%). Based on a minimum of 5 replicate injections of both standard solutions the relative difference Δ_rd_ is calculated as
(9)$$\Delta_{\mathrm{rd}}={\bar{C}}_{\mathrm{diff}}\%=100\times \frac{{\bar{C}}_{\mathrm{new}}-{\bar{C}}_{\mathrm{old}}}{C_{\mathrm{new}}}$$

If the pooled relative standard deviation of the responses is above the critical decision line we cannot state with 95% probability that the relative difference is within the acceptance criterion (10%). Further on, the figure indicates that |Δ_rd_| is inversely proportional to CV_p_. The closer the |Δ_rd_| to 10% the smaller the CV_p_ must be to verify compliance with the ≤10% criterion.

Since the typical repeatability CV of replicate injections into LC-MS/MS is about 2–2.5%, the maximum difference between the two standard solutions which can be stated ‘equivalent’ is about 7.67% and 7.09%, respectively. The calculation with Equation (10) is shown in Table 8.

The pooled CV_P_ is calculated as:(10)$$u{\mathrm{CV}}_{P}=\sqrt{\left({\mathrm{CV}}_{A1}^{2}+{\mathrm{CV}}_{A2}^{2}\right)/2}$$

The corresponding Δ_rd_ and CV_P_ values are plotted on Figure 6. It can be seen that for the B standard solutions the CV_p_ is above the critical line. Consequently, according to the TOST calculation, we cannot state with 95% probability that the difference between B1 and B2 standard solution is ≤10%.

### 2.7. Stability of Analytes

The pH of the plant fluids, enzymes released during the cutting, chopping of sample material can decompose sensitive analytes [87]. The analytes remaining in the final extract are also influenced by their physical−chemical properties, the temperature of comminution and mass of the laboratory sample. The disappearance of captan and dithiocarbamate residues during sample comminution was already observed in the middle of the 70s [104]. Hill reported the decomposition of chlorothalonil and phthalimide type of compounds especially in lettuce and onion [86]. The procedure for the determination of the stability of analytes was reported in the case of tomato, lettuce and maize [105]. It was found that buprofezin and chlorpyrifos did not decompose in the tested matrices at ambient temperature either. Their recoveries were well reproducible and close to 100%, therefore they can be used as reference compounds for assessing the stability of other analytes by comparing their residue concentrations surviving after comminution. Since the recovery tests performed with spiked test portions before extraction do not reveal any information on the stability of analytes, the stability tests should be executed as part of the method validation or performance verification for the new analyte matrix combinations with surface-treated sample material.

The stability test is practically the same as the procedural recovery. Its performance briefly described hereunder:(a)Take about two or more kg of the crop in which the stability of analytes will be tested.(b)Prepare the portion of the commodity to be analyzed according to Codex CAC/GL 41-1993 [81] from the whole laboratory sample.(c)Use approximately half of the sample matrix for the stability test and the remaining part for the recovery tests performed with spiked test portions as usual.(d)Prepare analytical standard mixture of exactly known concentration of compounds to be tested together with buprofezin (Bu) and chlorpyrifos (Ch) at well detectable concentrations keeping in mind the total mass of the sub-sample to be processed. The number of pesticides or metabolites included in the mixture is limited only by the capability of the chromatographic separation and detection system.(e)Take about 1/3 of the part of the laboratory sample (e.g., 3–4 units out of 10 fruits) obtained in step 3;(f)Treat the surface of the selected portion applying either Hamilton syringe for carefully spreading the standard mixture (step 4) on the surface of the crops or injecting the standard solution into the flesh of the fruit [106]. Use liquid dispenser to treat leafy vegetables or small-size crops. Perform the treatment in a fume cupboard over a tray with filter paper which can absorb the runoff. The exact amount of standard mixture that remains on the crop surface need not be known as the concentration ratios of the reference and test compounds will be calculated.(g)Steps for the treatment of the surface of tomatoes:
Place the surface-treated portions into the chopper together with the remaining 2/3 part of the sample and comminute the whole matrix. By this way you represent a potentially worst-case scenario for testing the efficiency of sample processing and determination of CV_L_ at the same time as testing the stability of analytes. The test may be performed both at ambient temperature and under cryogenic conditions applying dry ice or liquid nitrogen following the normal procedure applied in the laboratory.Verify the efficiency of comminution with a Petri dish test (See Section 2.3). Continue the process until an acceptable particle size distribution is obtained. Note that a lengthy process may increase the decomposition.Remove test portions from the comminuted matrix according to the normal procedure of the laboratory, but preferably from≥ 10 different positions.(h)Using the remaining part of the test material, perform the recovery test as usual by spiking the selected test portion with the standard mixture. Calculate the recovery for each compound.(i)Determine the concentration of survived residues from the surface-treated material and their recovery from spike test portions with the method to be applied.(j)Perform the test in ≥5 replicates.

#### Calculation of the Stability of Test Compounds

The measured concentrations of survived reference compounds are C_ch_ and C_Bu_. Average recoveries of chlorpyrifos and buprofezin from spiked test portion are denoted as ${\bar{\sigma }}_{Ch}$ and: ${\bar{\sigma }}_{Bu}$, respectively. The measured concentrations of the ‘i^th^’ test compound from surface treatment is C_i_. Average recovery of compound ‘i’ from the spiked test portion: ${\bar{\sigma }}_{i}$_._ Surviving residues are calculated with the average recoveries for each replicate test portion separately because the concentrations present in the test portions are different due to the inhomogeneity of the comminuted material (the efficiency of comminution). The survived portion of compound ‘i’ is calculated from the first test portion as:(11)$$\varphi_{\mathrm{iCh}1}=\frac{C_{i1}\times {\bar{\rho }}_{\mathrm{Ch}}}{C_{\mathrm{Ch}1}\times {\bar{\rho }}_{i}};\;\varphi_{\mathrm{iBu}1}=\frac{C_{i1}\times {\bar{\rho }}_{\mathrm{Bu}}}{C_{\mathrm{Ch}1}\times {\bar{\rho }}_{i}}$$

The first estimate of the survived portion (${\bar{\varphi }}_{i1})$ is calculated as the average obtained from the comparison with chlorpyrifos (Φ_iCh1_) and buprofezin (Φ_iBu1_). The stability of an analyte is characterized with the estimated grand average of survived portions of the ‘i^th^’ compound obtained from the n replicate measurements:(12)$${\overset{=}{\varphi }}_{i}=\frac{\sum_{1}^{n}{\bar{\varphi }}_{i}}{n}$$

The numerical calculations are illustrated in Table 9, Table 10 and Table 11. The surviving proportions were calculated with Equations (11) and (12). It is pointed out that each test portion was analyzed by different analysts taking part in one of our international training workshops. The test mixture used contained 17 pesticide active substances. The participants got acquainted with the QuEChERS method during the workshop and they had not used it before. Consequently, better reproducibility can be expected with the staff having experience with the method and working in their own laboratory. The results indicated the within-laboratory reproducibility of the analyses phase (CV_A_) and whole process of determination of pesticide residues (CV_L_). The summary of results is given in Table 9.

Comparing the CV_L_ values obtained with the analyses of 1 g and 10 g test portions clearly indicates the effect of test portion size on the reproducibility of the results which is about two times higher for the 1 g portion than the 10 g portion. The corresponding CV_A_ values (0.015 and 0.014) for buprofezin and chlorpyrifos do not show any dependence from the test portion size. It is not surprising because they reflect the repeatability of the procedure from the point of spiking of the test portions. The CV_Sp_ values, calculated with the rearranged Equation (2):(13)$${\mathrm{CV}}_{\mathrm{Sp}}=\sqrt{{\mathrm{CV}}_{L}^{2}-{\mathrm{CV}}_{A}^{2}}$$
are practically the same as the CV_L_ because the CV_A_ is much smaller. The average CV_Sp1_:CV_Sp10_ ratio is about two which is smaller than that predicted with Equation (7) indicating that the latter provides only an approximate tendency. These results underline the importance of regular testing the reproducibility of the whole pesticide residue determination process that can be done most conveniently with the reanalysis of retained test portions described hereunder.

### 2.8. Determination and Demonstration of within-Laboratory Reproducibility

According to various guidance documents [62,63,64] the precision of the analysis steps (CV_A_) should be determined with recovery tests. The individual recoveries are affected by the random and systematic errors. The sum of systematic errors is indicated by the average recovery, and the standard deviation of individual recovery values reflects the uncertainty (precision) of the measurements. Where the individual recoveries are within the 60–140% default range [62] and the average recoveries obtained for individual analyte sample matrix combinations are statistically not different (e.g., based on Grubb’s outlier test [102] the average recovery and the pooled CV_A_ (CV_AP_) can be calculated [107] and used for describing initially the performance of the method.
(14)$${\mathrm{CV}}_{\mathrm{AP}}=\sqrt{\frac{\sum^{\;}{\mathrm{df}}_{i}{\mathrm{CV}}_{\mathrm{Ai}}^{2}}{\sum^{\;}{\mathrm{df}}_{i}}}$$

Point to note: when the Grubb’s test is applied: there are several websites offering the critical values for the test. To obtain a correct outcome the critical values should be selected for a two-sided test, as given by ISO 5725 [102]. The initial estimate of method performance, based on a limited number of tests, should be verified, or refined, if necessary, during the ongoing performance verification that requires testing the recovery in each analytical batch. Keeping in mind the wide scope of the methods, covering often over 300 residues, it would not be practical to include all of them in every batch. It is recommended to test at least 10% of analytes (minimum five) included in the scope of the method in the rolling program together with various representative commodities from different commodity groups [62]. Consequently, hundreds of recovery values are generally generated in a laboratory monitoring pesticide residues. Each recovery value obtained on different days provides one estimate for the precision (relative uncertainty) of the results under within-laboratory reproducibility conditions. It is usually assumed that the random error of analytical results conforms to normal distribution because the total error is made up of the combination of small independent random errors arising at the various stages of an analytical procedure [108].

Assuming normal distribution, we can expect that the individual recovery values vary around the average (µ). Provided that the determination process is under “statistical control” 95% and 99.7% of the recovery values should be within the average (µ) ± 1.96sd and µ ± 3sd intervals (sd = standard deviation). Consequently, the control chart for individual recoveries is constructed based on the initial method validation data. The upper (UWL) and lower (LWL) warning limits encompass the µ ± 2sd range, whereas the corresponding action limits (UAL, LAL) are at µ ± 3sd. Since the probability for falling outside one of the action limits is very small (0.15%) such a situation would require immediate action by the operators. ISO 17025-2017 recommends preparing control charts to record the results in such a way that trends are detectable [109].

The original QuEChERS method has been used [52] with no or minor modifications in combination with GG-MS/MS and LC-MS/MS detection in one of our laboratories. The initial in-house validation of the method with representative analytes and sample materials resulted in an average recovery of 91.7% with a ‘within-laboratory reproducibility’ CV_AR_ = 9.6%. As part of the regular internal quality control 2354 recovery tests were performed at 0.01 mg/kg and 0.05 mg/kg levels altogether with 302 pesticide residues during the previous four months. The sample matrices included fruits and vegetables of high-water content such as apple, carrot, cucumber, eggplant, dragon fruit, grape, longan, mango, onion, orange, and sweet and chili pepper. Control charts were constructed for the selected groups of pesticides that were tested together. One example is shown in Figure 7, indicating only the results of the first 15 testing days with a limited number of pesticide residues to enable the graphical presentation and visual evaluation of the data.

Figure 8 shows that the recoveries were within the warning limits (74 and 109) of the randomly selected pesticides, though their distribution is not symmetrical, without displaying any clear tendency. In view of the size limitations of control charts, the periodic evaluation of a great number of recovery data (e.g., 2354 recoveries for the tested 302 compounds) can be better done based on their relative frequency diagram shown in Figure 9. The calculation can be easily done with Excel and has no size limitations. 

Figure 8 indicates that the highest recovery (117.5%) is within the upper action limit (UAL) (120%) and the lowest recovery is 70% well above the 65% lower action limit. Moreover, the mean recovery (91.7%) determined during the validation of the method is encompassed by the most frequently occurring 90% and 95% recoveries within the warning limits. The results of the 2354 recovery tests confirm that the tested 302 substances can be determined in fruits and vegetable samples with the typical performance parameters established during the validation of the method.

The long-term within-laboratory reproducibility (CV_L_) of the residue determination process, which incorporates the contribution of subsampling, sample homogenization and analyses (Table 2), can be most conveniently determined [65] with the reanalyzes of the retained test portion that is also recommended by ISO17025:2017 as an internal quality control action [109]. The retained test portions must be obtained from samples containing incurred residues to demonstrate the efficiency of comminution. Analysts should be aware that only the CV_L_ can indicate the performance of the whole determination process and not the CV_A_. Therefore, CV_L_ should be determined regularly for each type of commodity as part of the internal quality control plan of the laboratory. For performing the reanalyzes of retained test portions, prepare 10–15 test portions from each sample. If residues are detected, keep the test portions for further analyses. If no residue is detected a few test portions may be kept for preparing matrix-matched calibration solutions. The remaining test portions can be discarded. In due course of the regular analyses of various samples, a retained test portion should be included in the analytical batch and blindly reanalyzed. The results should be recorded in the format shown as an example in Table 12.

The results of the reanalyzes of retained test portions may be evaluated based on the standard deviation of the difference of the two measurements made on “closely similar materials containing residues fairly close in amount present” [110]. The number of sample portion pairs analyzed should be ≥5 to obtain a realistic estimate for the CV_L_. Since the average residues of the original and retained test portions are different, their relative difference should be used for the estimation of CV_L_.
(15)$${\mathrm{CV}}_{L}=\sqrt{\frac{\sum^{\;}R_{\Delta i}^{2}}{2n}}$$
where R_ΔI _= 2(R_i1_ − R_i2_)/(R_i1_ + R_i2_), R_i1_ and R_i2_ are the residues obtained from the analyses of the i^th^ test portions and n is the number of test portion pairs. Assuming that only random error affects the duplicate measurements, their average must be zero, thus the degree of freedom is equal to ‘n’, the number of measurement pairs. Alternately, the range statistics [111] can be used for the estimation of CV_L_ that does not assume the above-mentioned preconditions specified by Youden. For the i^th^ measurement pairs the CV_Ri_ is calculated with Equation (16). The d_2_ for two replicate measurements is 1.128.
(16)$${\mathrm{CV}}_{\mathrm{Ri}}=\frac{R_{\max}-R_{\min}}{\bar{R}\times d_{2}}$$

The CV_L_ is calculated from ‘n’ test option pairs with pooling the CV_Ri_ values [112]):(17)$${\mathrm{CV}}_{L}=\sqrt{\frac{\sum^{\;}{\mathrm{CV}}_{\mathrm{Ri}}^{2}}{n}}$$

The degree of freedom for the corresponding standard deviations [sd = CV_L_ × R] of the measured residues (R) is equal to ‘n’. The two estimates of CV_L_ with Equations (15) (0.1283) and (17) (0.1608) are slightly, but statistically not significantly, different. We recommend using the larger CV_L_ to avoid underestimating the long term within-laboratory reproducibility of the residue determination process.

### 2.9. Chromatographic Determination of Residues

The gas and liquid chromatographic separation and MS detection conditions are generally well described in the publications often following the guidance given by SANTE/11312/2021, SANTE/2020/12830, USFDA documents [62,63,64]. However, there are a few points that should be considered when the chromatographic conditions are characterized.

The reported LOD values or reporting limits should always be checked at the beginning and at the end of the analytical batch of sample extracts for all targeted analytes preferably in blank sample extract, because loading the column with coextracted materials may change the resolution of the column and or shift the retention times as illustrated in Figure 9 and Figure 10. This is especially important in the case of screening methods for unknown pesticide residues in monitoring programs.

The inertness and satisfactory operating conditions of gas chromatographic columns can be improved by applying the so-called analyte protectants [112,113] A critical review and re-assessment of analyte protectants in gas chromatography was published by Rodríguez-Ramos [114].

**Figure 9 molecules-28-00954-f009:**
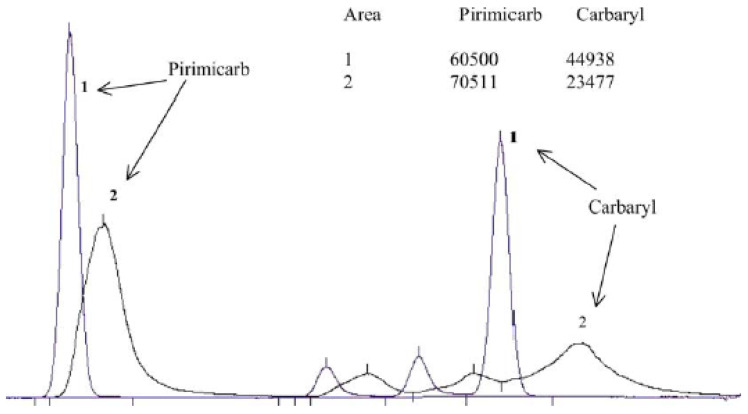
Deterioration of chromatographic peak shape after injecting 40 sample extracts (5 g sample/mL). Taken with permission form [115].

**Figure 10 molecules-28-00954-f010:**
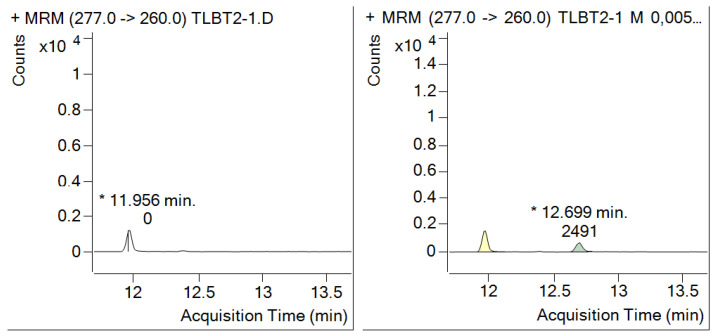
Shifting the retention time from 11.956 to 12.699 and changing the shape of the response of fenitrothion at the beginning and at the end of the analytical batch. Note the shapes of peaks obtained after injection of 0.005 μg/mL considered to be the LOD. MRM: multi reaction monitoring mode, TLBT2-1: sample identifier.

The data analyses reports provided by the software should not be viewed as a ‘black box’ and accept it without verification of its correctness. The modern data analyzers (e.g., Aglient Mass Hunter) usually offer six different curve fit types (linear, quadratic, power, first order ln, second order ln, and average of response factors), four possible choices for the origin (ignore, include, force, blank offset), and seven for weighing (none, 1/x, 1/x, 1/y, 1/y^2^, Log, 1/sd^2^). The reported results can be quite different depending on which integration options are selected. Attention is also required to assess the number of disabled points and the reported confidence limits of the slope and intercept of the regression equations. For instance, where three out of six calibration points are disabled the predicted analyte concentration should be critically considered, and possibly additional calibration injections should be made.

Chromatograms must be inspected by the analyst and the actual baseline fit examined and adjusted, if necessary. The response of the suspected peaks should always be checked to verify that the ion(s) acquisition includes the whole peek(s) and their integration is correct (Figure 11).

For multi-level calibration the standard concentrations should be equidistantly distributed over the calibrated range. Figure 12 illustrates a frequently applied questionable practice where four calibration points [ng/mL] were in the first 1/10 part of the calibrated range (0.05, 0.1, 0.5, 1, 5, 10 μg/kg).

Such a calibration program type is only justified where analytes potentially present at low concentrations are looked for in screening analyses.

It should be recognized that the correlation coefficient (r) or the coefficient of determination (R^2^) provides information only on the linearity of the calibration but does not characterize the quality of the calibration. It can be assessed based on confidence intervals, calculated by those of the data processing software for the slope and intercept of the regression line or from the standard deviation of the relative residuals. The latter parameters should also be reported together with ‘r’ or R^2^.

Figure 13 and Figure 14 show calibration charts with confidence and tolerance intervals around the linear regression line obtained with 1/x weighting [116]. Note that the R^2^ values indicating the linear fit are practically the same, but the standard deviation of relative residuals (Sd_rr_) indicating the scatter of the responses around the regression line as well as the width of the confidence and tolerance intervals are substantially different. The confidence intervals around the regression line are strongly influenced by the number of standard injections (not shown in the figure).

The regression residual $\Delta y_{i}$ describes the vertical distance of measured responses from the regression curve according to:(18)$$\Delta_{\mathrm{yi}}=y_{i}-{\hat{y}}_{i};\;\Delta_{\mathrm{yirel}}=\frac{\Delta_{\mathrm{yi}}}{{\hat{y}}_{i}}$$

The standard deviation of the relative residuals is calculated as:(19)$$Sd_{\Delta y/\hat{y}}=\sqrt{\frac{\langle \Delta_{yirel}-{\bar{\Delta }}_{rel}{}^{2}\rangle }{n-2}}=Sd_{rr}$$

When each reference material is measured *k* times, the number of degrees of freedom is (nk–2).

Nonetheless the R^2^ values are practically the same, the Sd_rr_ values indicating the large difference in the confidence/tolerance intervals in Figure 13 and Figure 14. Table 13 shows further examples from our practice underlying the fact that the R^2^ is not a proper indicator of the accuracy of the calibration [65]. Our experience suggests that for accurate calibration the Sd_rr_ should be <0.1 (10%). The Codex quality control guidelines suggest accepting a maximum of 20% relative residuals (30% near the instrument LOQ) [71].

## 3. Discussion

The monitoring programs are conducted around the world including large number of samples to provide data for carrying out:dietary exposure assessment of consumers;evaluating the residue levels and their compliance with national or international maximum residue limits or guidance values;assessing the contamination of the environment;providing the basis for the necessary corrective actions if the residues exceed the reasonably expectable levels in the treated crops.

Each analysis may have significant consequences. Therefore, the results should be representative and defendable even in legal proceedings. Analysts must be aware of their responsibilities and the fact that their credibility could be at stake. They should be able to verify the correctness of their measurements with documented evidence.

The international standards and guidelines provide the frame and acceptable performance criteria for performing the pesticide residue analytical measurements. They would facilitate obtaining accurate, defendable results only if the laboratory operations are performed by staff members (from the top manager, who has the key role, to each member) who are aware of their own responsibility and are working in coordination with each other.

It is not sufficient to validate our methods or test the performance of already validated methods once. The laboratories should establish their own internal quality control programs to be used daily for ensuring that their methods satisfy the specified performance characteristics when applied for instance to screen over several hundreds of analytes in samples of unknown origin or to test the residues in commodities before export.

The provisions of guidance documents should be fulfilled bearing in mind that the priorities of internal quality control are in order: (1) good analytical practice; (2) good science; (3) minimum bureaucracy; (4) facilitating reliability and (5) efficiency. The quality assurance/quality control (QA/QC)should only be an appropriate proportion of the activities related to the analyses of samples and reporting of the results.

Keeping in mind the above priorities, we emphasize that it is not sufficient to report the recoveries obtained with spiked test portions, the linearity of calibration, detection conditions, and confirmation of the identity of substances. In addition, we propose checking and preferably briefly reporting, for instance, the validity of samples considering the parameters that can be verified in the laboratory, accuracy of analytical standards, stability of analytes during the laboratory operations, quality of calibration characterized with the relative residuals or their standard deviation, and the reproducibility relative standard deviation of the measured residues.

Moreover, the selection of the parts of samples and the composition of the residues to be determined should always be matched with the objectives of the work.

It is advisable to take part regularly in proficiency tests that provide a means of objectively evaluating and demonstrating the accuracy and reliability of our measurements. Critical review of the Z-scores and identification of the sources of the potential errors can help to improve the technical operation standard of the laboratory. However, participating in proficiency tests does not replace the regular and rigorous internal quality control actions.

Finally, reliable results on which regulatory decisions are based can be expected only from well-trained analysts whose knowledge should be regularly updated to fully utilize the advantages of the high-performance instruments and benefit from the rapidly expanding methodical experience gained by other laboratories through the analyses of a great variety of samples.

## Figures and Tables

**Figure 1 molecules-28-00954-f001:**
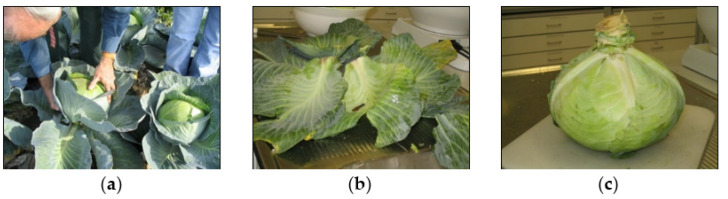
Phases of preparation of head cabbage for analysis: (**a**) collecting head cabbage; (**b**) removing outer leaves; (**c**) obtaining portion of commodity to be analysed after cutting off the stalk.

**Figure 2 molecules-28-00954-f002:**
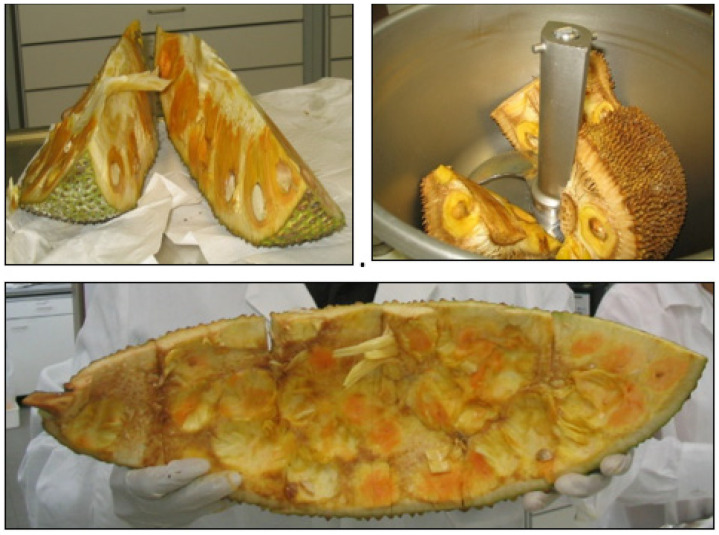
Processing of jackfruit: top left: cutting wedge-shaped section from jackfruit; top right: comminution of the whole fruit; bottom: peel remaining after removal of edible part.

**Figure 3 molecules-28-00954-f003:**
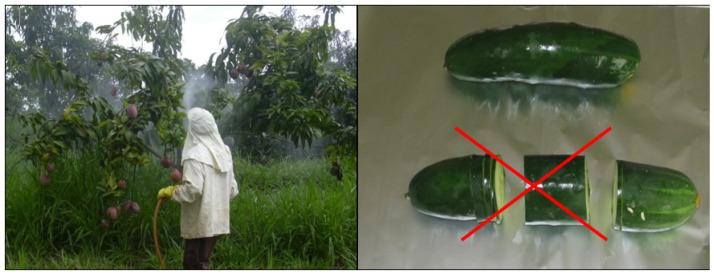
Distribution of residues in/on crop units: Left: Residues concentrate on the bottom part of fruits hanging on the trees; Right: Cutting the middle of cucumber leads to biased result. Slices should never be cut for subsampling as emphasized by the crossing red lines.

**Figure 4 molecules-28-00954-f004:**
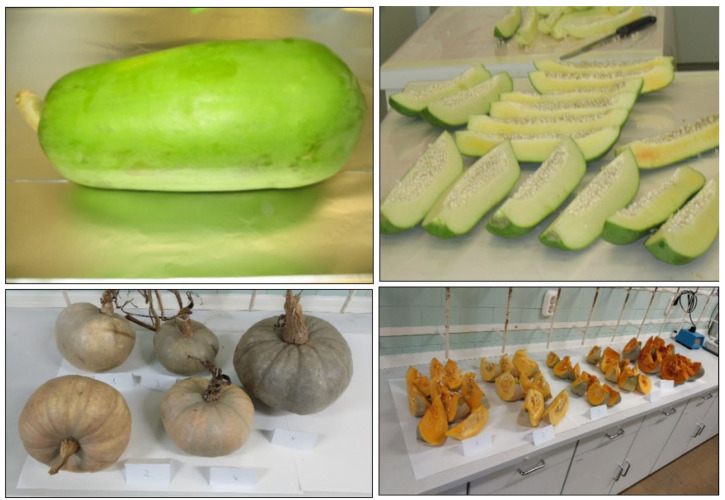
Cutting wedge-shaped portions of large fruits. Top left: papaya fruit; Top right: papaya subsamples; Bottom left: winter squash; Bottom right: sections of winter squash. Note: that the cut portions from each fruit should be kept in separate groups until taking the required number of pieces from each group for comminution.

**Figure 5 molecules-28-00954-f005:**
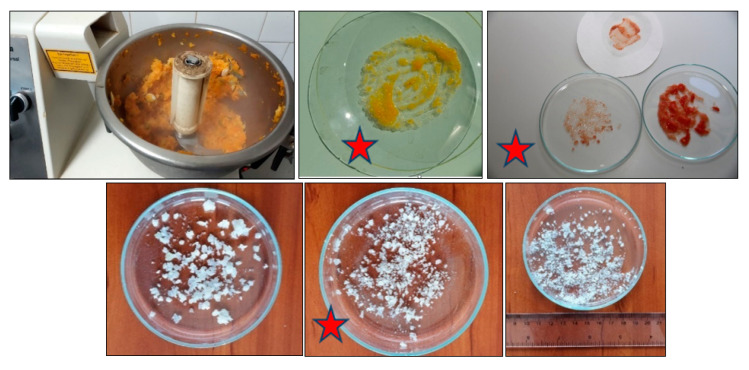
Top row left Blending winter squash; top center and right: tomato homogenates on filter paper and Petri dish; second row: cabbage leaves homogenized to different particle sizes. The star marks the acceptable particle size distribution.

**Figure 6 molecules-28-00954-f006:**
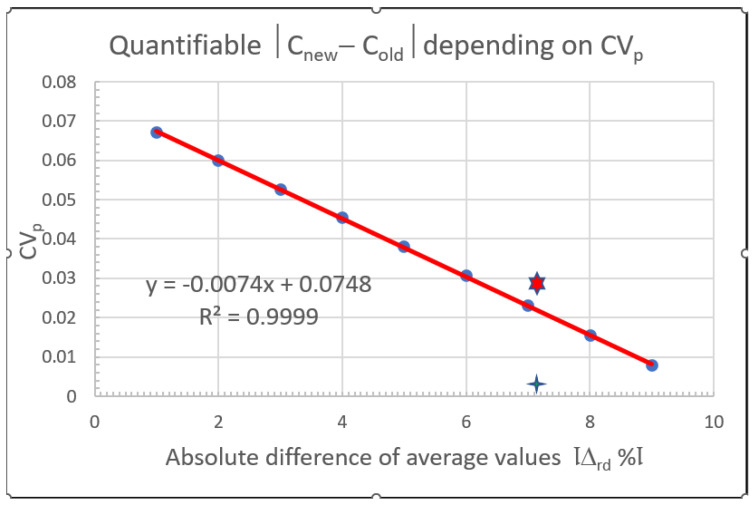
Relation of the absolute value of Δrd and maximum CV_p_ that can be used to verify that Δrd ≤ Θ = 10% at 95% probability level if n_1_ = n_2_ = 5. The symbols 
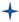
 and 
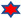
 indicate that the acceptance criterion (Δrd ≤ 10%) can or cannot be confirmed, respectively. Taken with permission from [103].

**Figure 7 molecules-28-00954-f007:**
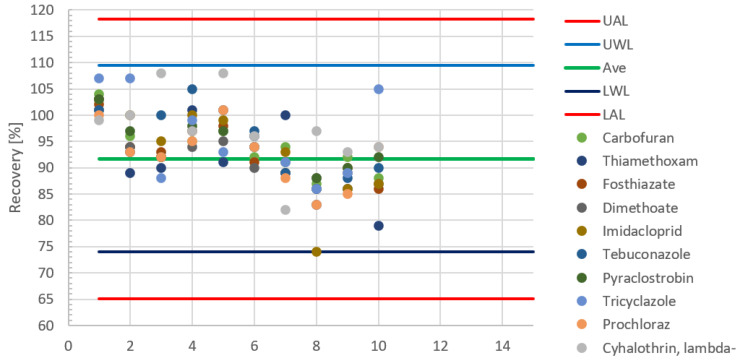
Example for the control chart demonstrating the within-laboratory reproducibility of the analyses phase of the determination of ten pesticide residues. Action (AL) and warning limits (WL) are indicated with red and blue lines, and the green line shows the average recovery.

**Figure 8 molecules-28-00954-f008:**
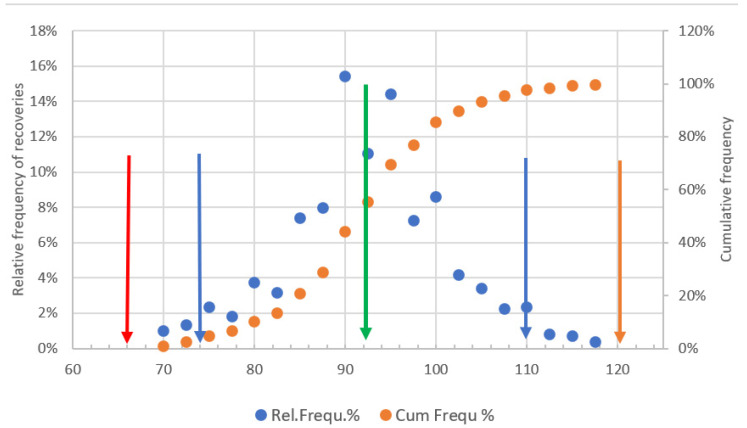
Relative and cumulative frequency distribution of 2354 recovery data obtained with 302 pesticide residues in commodities of high-water content during a four-month period. Red and blue arrows indicate the action and warning limits, respectively. Green arrow indicates the average recovery.

**Figure 11 molecules-28-00954-f011:**
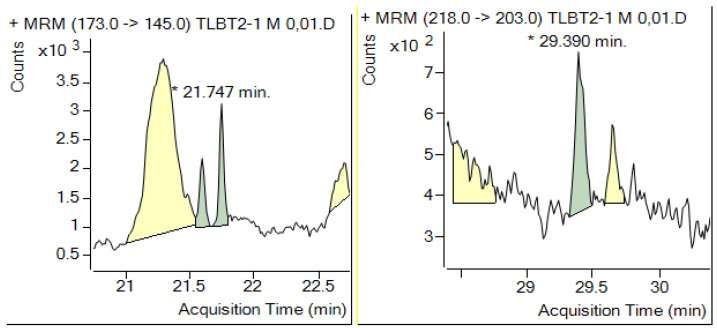
Examples for verification of correct identification of suspected peaks of propiconazole and indoxacarb in left and right pictures. MRM: multi reaction monitoring; TLBT2-1: sample identifier.

**Figure 12 molecules-28-00954-f012:**
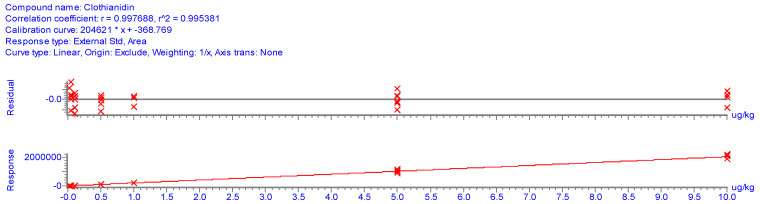
Improper selection of calibration points that should be equidistantly distributed. ∗ indicates the position of the response obtained with the injection of standard solutions.

**Figure 13 molecules-28-00954-f013:**
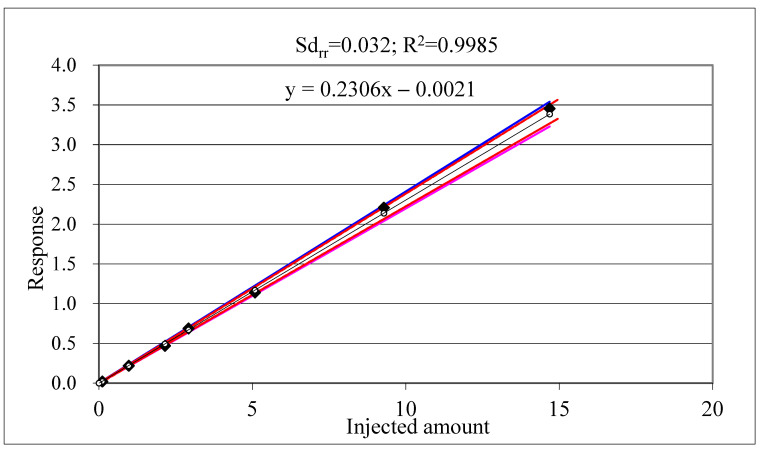
Terbuthylazine calibration charts. Blue lines indicate the confidence intervals, the red lines the tolerance intervals around the regression line that were calculated applying the approximation recommended by Miller and Miller [116].

**Figure 14 molecules-28-00954-f014:**
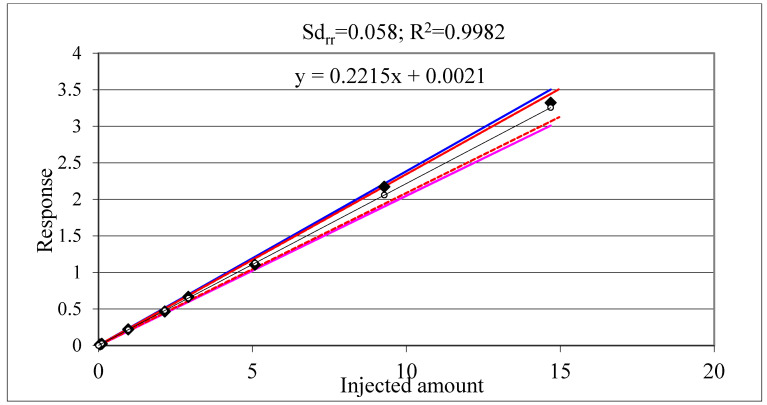
Terbuthylazine calibration charts. Blue lines indicate the confidence intervals, the red lines the tolerance intervals around the regression line that were calculated applying the approximation recommended by Miller and Miller [116].

**Table 1 molecules-28-00954-t001:** Examples for the sources of errors in the results of pesticide residues determination.

Potential Sources of Errors		
Operation	Random	Systematic
Sampling	Sample size; heterogeneous distribution of analyte; varying temperature during shipping and storage	Sampling target selection; sampling plan and method; degradation, evaporation of analyte; contamination of the sample; mislabeling
Selection of the portion of commodity to be analyzed	Inconsistent preparation of sample portion	Wrong part of the sample selected for extraction,
Sample size reduction, subsampling	Subsample does not represent the composition of the laboratory sample	primary samples are not proportionally represented
Comminution of selected sample portions	Particle size distribution in the homogenate; varying temperature and duration of comminution	Decomposition, evaporation of analytes
Test portion selection	Test portion does not represent the comminuted sample matrix	
Extraction	Varying intensity and temperature of extraction	Efficiency of extraction
Clean-up	Variation in the composition (e.g., water, fat, and sugar content) of sample materials;	Loss of analyte
Qualitative/quantitative determination of residues	Changing the retention time—shifting mass acquisition window; linearity and confidence intervals of calibration	Deviation from residue definition; missing analytes present in targeted or non-targeted analyses; high LOD; inaccurate standard solutions; matrix effect

**Table 2 molecules-28-00954-t002:** Change of CV_Sp_ as a function of the test portion mass.

CV_Sp_		
m_L_ [g]	>5000	1000
T_p_ [g]		
1	0.387	0.387
2	0.274	0.274
5	0.173	0.173
10	0.122	0.122
15	0.100	0.099
25	0.077	0.076

**Table 3 molecules-28-00954-t003:** Different residue definitions for flupyradifurone.

Flupyradifurone [4-[(6-chloro-3-pyridylmethyl)(2,2-difluoroethyl)amino]furan2(5H)-one	
definition of the residue (for compliance with MRLs) for plant commodities	flupyradifurone
definition of the residue (for dietary risk assessment) for plant commodities	sum of flupyradifurone, difluoroacetic acid and 6-chloronicotinic acid, expressed as parent equivalents

**Table 4 molecules-28-00954-t004:** Different residue definitions for fluxapyroxad.

Fluxapyroxad [3-(difluoromethyl)-1-methyl-N-(3′,4′,5′-trifluoro [1,1′-biphenyl]-2-yl)-1H-pyrazole-4-carboxamide]	
definition of the residue (for compliance with the MRL for plant and animal commodities)	fluxapyroxad
definition of the residue for estimation of dietary intake for plant commodities	sum of fluxapyroxad and 3-(difluoromethyl)-N-(3′,4′,5′-trifluoro[1,1′-biphenyl]-2-yl)-1H-pyrazole-4-carboxamide (M700F008) and 3-(difluoromethyl)- 1-(ß-D-glucopyranosyl)-N-(3′,4′,5′-triflurobipheny-2-yl)-1Hpyrzaole-4-carboxamide (M700F048) and expressed as parent equivalents
for estimation of dietary intake for animal commodities	sum of fluxapyroxad and 3-(difluoromethyl)-N-(3′,4′,5′-trifluoro[1,1′-biphenyl]-2-yl)-1H-pyrazole-4-carboxamide (M700F008) expressed as parent equivalents; the residue is fat soluble

**Table 5 molecules-28-00954-t005:** Summary of results of EU-RT-FV-17 ^a^.

	Bupirimate	Carbendazim	Diazinon	Difenoconazole	Diflubenzuron	Methoxyfenozide	Pendimethalin	Permethrin	Spinosad	Thiabendazole	Trifloxystrobin
Certified conc. mg/L	5.00	5.00	5.04	18.99	18.96	14.95	4.97	15.05	15.03	19.04	19.00
No. Lab	33	31	36	34	25	30	35	32	30	32	33
Accurate	2	1	0	0	0	0	0	0	0	0	0
Rel dif.% Min	−74.2	−86.6	−41.5	−43.7	−40.9	−51.5	−59.8	−54.1	−36.1	−32.5	−36.0
Rel dif.% Max ^b^	40.0	164	202	36.9	129	107	28.0	73.4	91.0	116	118
No ≥ 10%	17	23	18	19	16	18	19	23	19	16	19

^a^: Courtesy of Carmen Ferrer Amate; ^b^: rounded to 3 digits; No. Number of laboratories: reported result; Accurate: Certified = reported; No ≥ 10%: number of laboratories reported >10% rel. difference.

**Table 6 molecules-28-00954-t006:** Example of reproducibility of filling A-grade volumetric flasks.

Vol. Flasks	Specification	CV_T_	CV_Rfil_	CV_Rexp_
25 mL	±0.03 mL	4.899 × 10^−4^	7.30 × 10^−3^	7.32 × 10^−3^
50 mL	±0.05 mL	4.082 × 10^−4^	7.59 × 10^−4^	8.61 × 10^−4^

**Table 7 molecules-28-00954-t007:** Example for the reproducibility of preparation of analytical standard solutions.

C_0_ mg/mL	CV_R_	Effective Concentration			Deviation ^1^ from C_0_ [%]		
		C_min_	C_ave_	C_max_	Min	Average	Max
1	0.0079	0.9840	1.0044	1.0084	−1.6%	0.44	0.84
0.005	0.0086	0.0050	0.0052	0.0053	0%	4.00	6.00
0.001	0.0103	0.0010	0.0011	0.0011	0%	10.00	10.0

^1^: Deviation of the effective concentration from the nominal concentration [C_0_].

**Table 8 molecules-28-00954-t008:** Testing the difference in nominal concentrations of analytical standard solution.

	Standard A1	Standard A2	Standard B1	Standard B2
	121315	112823	123453	114811.3
	121525	112813	131282	122092.3
	121310	113000	123456	114814.1
	121401	113121	124356	115651.1
	121392	112802	123451	114809.4
Ave	121388.6	112911.8	125199.6	116435.6
Δrd	7.2%		7.3%	
CV	0.000718	0.001262	0.027337	0.027337
CVp	0.001027		0.027337	
	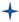		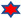	

The symbols 
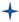
 and 
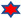
 indicate that the acceptance criterion (Δ_rd_ ≤ 10%) can or cannot be confirmed, respectively.

**Table 9 molecules-28-00954-t009:** Results of stability test performed with 1 g test portions at ambient temperature.

Recovery Tests with 0.2 mg/kg Spike						Survived Residues [mg/kg]			
	Residues Measured [mg/kg]								
	Bu	Ch	Etri	Etox		Bu	Ch	Etri	Etox
	0.177	0.165	0.137	0.176		0.161	0.157	0.122	0.139
	0.186	0.182	0.156	0.176		0.173	0.180	0.117	0.165
	0.204	0.183	0.151	0.153		0.129	0.116	0.100	0.133
	0.178	0.152	0.164	0.170		0.142	0.128	0.108	0.132
	0.169	0.169	0.150	0.164		0.135	0.137	0.106	0.125
$\bar{\rho }$	0.913	0.852	0.758	0.840	$\bar{\rho }$	0.207	0.231	0.176	0.217
CV_A_	0.015	0.015	0.013	0.011	CV_L_	0.698	0.937	0.908	0.885

Notes: Ch: chlorpyrifos, Etri: etridiazole; Etox: etoxazole; $\bar{\rho }$: average recovery. The Table shows rounded values, but the calculations were performed with four-digit numbers.

**Table 10 molecules-28-00954-t010:** Proportion of survived residues based on 1 g test portion ^1^.

	Etridiazole with Bu	Etridiazole with Ch	Etoxazole with Bu	Etoxazole with Ch
	0.523	0.502	0.662	0.636
	0.469	0.420	0.731	0.655
	0.533	0.555	0.788	0.819
	0.528	0.547	0.711	0.737
	0.542	0.500	0.709	0.654
φ	0.519	0.505	0.720	0.700
$\overset{=}{\varphi }$	0.512		0.710	

Note: ^1^: The proportions of survived residues were calculated applying both buprofezin (Bu) and chlorpyrifos (Ch) as stable reference compounds.

**Table 11 molecules-28-00954-t011:** Summary of recoveries, survived residues, CV_A_ and CV_L_ values obtained with the tests performed at ambient temperature during the training workshop.

Parameter	Bu	Ch	Etri	Etox
Spiking 1 g test portion (5 replicates)				
Average recovery	0.913	0.852	0.758	0.840
CV_A_	0.015	0.015	0.013	0.011
Spiking 10 g test portions (5 replicates)				
Average recovery	0.970	0.949	0.815	0.957
CV_A_	0.014	0.014	0.027	0.017
Extracting 1 g portion from surface-treated tomato				
Average survived [mg/kg]	0.148	0.143	0.111	0.139
CV_L_	0.123	0.175	0.080	0.110
CV_Sp_	0.122	0.175		
Extracting 10 g portion from surface-treated tomato				
Average survived [mg/kg]	0.146	0.146	0.135	0.145
CV_L_	0.055	0.098	0.082	0.125
CV_Sp_	0.053	0.097		

**Table 12 molecules-28-00954-t012:** Results of the analyses of retained test portions (example).

Test No ^1^	Original Test Portion				Retained Test Portion		
	Sample code	Date of anal. ^2^	Residue/commodity		Test portion Code ^3^	Date of anal.	Residue [mg/kg]
			Name	[mg/kg]			
1	M261	22 August 2022	Bupirimate/ orange	0.205	M261/1	29 August 2022	0.216
2					M261/2	6 September 2022	0.210
3					M261/3	14 September 2022	0.195
4	M283	15 September 2022	Lufenuron/ pepper	0.52	M283/1	22 September 2022	0.75
5					M283/2	26 September 2022	0.45
6					M283/3	3 October 2022	0.50
7					M283/4	10 October 2022	0.68

Notes: ^1^: The repeated tests can be performed at various time intervals after the first analysis. ^2^: Date of the first analysis of the sample. ^3^: Test portions retained form the sample at the time of the first analyses.

**Table 13 molecules-28-00954-t013:** Examples for the corresponding Sd_rr_ and R^2^ values.

Sd_rr_	R^2^
0.042	0.9937
0.061	0.9976
0.085	0.9988

## Data Availability

All data used are included in the manuscript.

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
