# Peer review of "Quality Control of Pesticide Residue Measurements and Evaluation of Their Results"

_molecules, 2023, doi:10.3390/molecules28030954_

Round 1

Reviewer 1 Report

For Figures 1, 2, 3 and 7: please expand the captions and elaborate briefly the content of the figure.

Line 20: Spell check ‘reside’ vs ‘residue’

Scientific literature is poorly conducted, needs to be updated, for instance the introduction, results and discussion. I recommend authors to consider some bioanalytical and bionsensing-based studies for pesticide detection (https: //doi.org/10.1016/j.teac.2022.e00184), this will help to enrich the literature. See below few important discoveries and novel sensors for pesticide analysis from food such as, fenitrothion (https: //doi.org/10.3390/ijms221910846), malathion (https: //doi.org/10.1016/j.biomaterials.2022.121617).

Line 498: spell check ‘absolut’

Line 499: Delete author name and year, keep reference number.

Authors could add the significance and key results into the conclusions sections. Need to elaborate conclusions and provide future directions/implications of the study.

Please signify the relation between the method of analysis and sampling procedures in conclusion and abstract.

Overall, the study is significant for scientific advancement of pesticide analysis and could be considered for publication after suggested revision.

Author Response

  1. For Figures 1, 2, 3 and 7: please expand the captions and elaborate briefly the content of the figure.

The legends of figures were expanded.

2. Line 20: Spell check ‘reside’ vs ‘residue’ done

3. 

Scientific literature is poorly conducted, needs to be updated, for instance the introduction, results and discussion. I recommend authors to consider some bioanalytical and bionsensing-based studies for pesticide detection (https: //doi.org/10.1016/j.teac.2022.e00184), this will help to enrich the literature. See below few important discoveries and novel sensors for pesticide analysis from food such as, fenitrothion (https: //doi.org/10.3390/ijms221910846), malathion (https: //doi.org/10.1016/j.biomaterials.2022.121617).

Answer:

We closely follow the development of detection techniques including biosensors. However, this is not a review article, and we did not aim to provide the full coverage of the very extensive literature. The publications referenced serve to provide examples for the background for emphasizing the need for the careful consideration of the control of the usually hidden errors in the determination process to assure that the results are accurate, and the quoted repeatability reproducibility relative standard deviations reflect the whole process.

Moreover, this article is prepared for the special issue of Molecules titled: "Chromatographic Analysis of Pesticide in Environmental and Food Samples” The introduction of the special issue states: “This Special Issue specifically focuses on the determination of pesticide residues at trace levels in environmental and food matrices by chromatographic techniques combined with mass spectrometry or other suitable detectors, including the development and validation of analytical methods as well as monitoring studies.”

Though bioanalytical methods and using biosensors are important options for the future detection of residues in very specific circumstances, such methods are not designed to compete with the multi-residue methods used for monitoring pesticide residues.  Therefore, they are outside of the scope of the special issue and our manuscript.

Line 498: spell check ‘absolut’ corrected

Line 499: Delete author name and year, keep reference number: corrected

Authors could add the significance and key results into the conclusions sections. Need to elaborate conclusions and provide future directions/implications of the study.

Answer:

The template for the preparation of manuscripts indicates:

Conclusions:: This section is not mandatory but can be added to the manuscript if the discussion is unusually long or complex

Consequently, all our conclusions are summarized in the first part of the Discussion (lines 918-946), and recommendations for future actions, and directions are included in the second part (.947-978).

Please signify the relation between the method of analysis and sampling procedures in conclusion and abstract.

Lines 20-23 of the abstract address your point:

The validity of the analytical results can be achieved by the implementation of suitable quality control protocols during sampling and determination of pesticide residues. 

Moreover, the Discussion includes advice to check the validity of sampling L. 964-966.

Reviewer 2 Report

This topic is interesting considering dietary exposure assessment of consumers to pesticide residues and for verifying the compliance of the residue concentrations in food with the national or international maximum residue limits. However, I suggest that the manuscript be carefully reviewed before being published.

Below I present my main comments and suggestions, which hopefully can help the author/s improve the paper.

1.Page 7 Figure 2: Should the effect of kernels be taken into the pretreatment of jackfruit?

2.Page 10 line 305: The addition of water to dry materials can seriously alter the weight of the real sample and thus affect the results. Please make sure this operation is correct.

3.Page 18 Figure 7: Referring to the author's previous statement, “Therefore, slices should never be cut from crop units”. The residue is also unevenly distributed in the tomatoes and whether it should not be cut in this way.

4.Page 30 Figure 14: Note the difference between "," and the decimal point. For example, Srr=0,032 should be Srr=0.032. Improve the icon details in the text, there are more basic errors.

5. The article makes extensive use of others' literature to show that many operations can affect CVSp, and does not offer its own insights or suggest the correct approach to address these effects. It is recommended that this section should be supplemented.

6. The author should write "Methods" and "results" separately to avoid confusion.

Author Response

1.Page 7 Figure 2: Should the effect of kernels be taken into the pretreatment of jackfruit?

Normally not.

Your comment draws our attention to be more specific. Therefore, we added a clarification in Lines 203-208:

The edible portion varies and for instance depends on the variety, maturity of the crop, and the local practices for its consumption. Therefore, the actual way of selecting the edible portions should be precisely described in the publications to enable the comparison of the results with other studies.

Further on we extended the sentence to make the process explicit (line 225) To prepare samples for the analyses of the edible portion, the peeling the fruits with inedible peel…

2.Page 10 line 305: The addition of water to dry materials can seriously alter the weight of the real sample and thus affect the results. Please make sure this operation is correct.

To avoid any potential misunderstanding a sentence is added in Lines 308-309: ‘The exact amount of added water shall be accounted for in reporting the residue concentrations.’

3.Page 18 Figure 7: Referring to the author's previous statement, “Therefore, slices should never be cut from crop units”. The residue is also unevenly distributed in the tomatoes and whether it should not be cut in this way.

The recommended method is relevant for large crops because all of them making up a composite sample cannot be placed in the generally used choppers. Tomato and other medium and small crops can usually be processed without cutting representative portions.

Figure 7 shows the surface treatment of tomatoes. In this case, all treated fruits are comminuted together with the untreated ones (g/I section) to provide the matrix for the evaluation of the stability of residues and testing the reproducibility of withdrawing small test portions for extraction.

4.Page 30 Figure 14: Note the difference between "," and the decimal point. For example, Srr=0,032 should be Srr=0.032. Improve the icon details in the text, there are more basic errors.

The coma was replaced by dot in the figures

  1. The article makes extensive use of others' literature to show that many operations can affect CVSp, and does not offer its own insights or suggest the correct approach to address these effects. It is recommended that this section should be supplemented.

We would like to emphasize that we prepared a concept paper that draws attention to the potential errors in the determination of pesticide residues that need to be eliminated or minimized to obtain accurate and defendable results.  

To assist the analysts in selecting appropriate IQC tests that are fit for their purpose, we present our internal quality control procedures (methods) as examples mostly published by ourselves earlier, and we do not provide the results obtained during the routine use of these procedures. To avoid any confusion, we deleted ‘and results’ from the heading of section 2. The examples represent our insight into the problem and the best practices as well as the recommended approaches.

  1. The author should write "Methods" and "results" separately to avoid confusion.

Please see the explanation given for comment 5.

Round 2

Reviewer 1 Report

The authors have not incorporated suggestions during the revision.

Moreover, several errors and grammatical issues are found to be introduced during revised submission. For example: there should a space between units and numbers, this is not followed consistently (Line 450 and 453). Moreover, the ml should be written scientifically correct as mL (SI unit).

Figure 7: is not meaningful, can be put in supplementary files or deleted.

Table 8: At the bottom two colored designs are included, but they have no meaning.

Line 38: world should not be capitalized.

Line 51: Sentence is grammatically wrong. It should be “authorities authorize use of pesticides only after the critical evaluation of their”

There several cases of sentence lacking coherency and structure. Grammar needs urgent attention.

 Only after careful revision the manuscript can be considered.

Author Response

Dear Reviewer,

Thank you very much for calling our attention to the parts of our manuscript that needs to be improved or corrected. After careful consideration of the general and specific comments the following actions were taken or an explanation was given for making no change.

Testing compliance with MRLs is one of the main objectives of the monitoring programs. The analysts should be aware of the process of the authorization of pesticides, establishing MRLs, and the different residue definitions for enforcement and risk assessment purposes to produce residue data that fit the purpose. The very brief summary in the introduction assists interested analysts in finding the right references for further information.

The quoted references to monitoring programs provide the background for the current situation and highlight the importance of applying for appropriate internal quality control programs and publishing its main results. Since this is not a review paper, we provide only examples to illustrate the current practice and do not attempt to provide full coverage of the monitoring results and methods used for the analyses of samples. In this regard, all references are relevant to the topic of the article.

This is not a report of a research program therefore strictly speaking the research design is not applicable.

The authors have not incorporated suggestions during the revision.

All your comments made in the first round had been carefully considered and the text of the manuscript was consequently modified or expanded.

It was also pointed out that the conclusions on the key results and suggestions for future directions had already been included in the discussion section.

Therefore, we believe that all your points were fully considered, and appropriate actions were taken.

Specific comments:

1. Moreover, several errors and grammatical issues are found to be introduced during revised submission. For example: there should a space between units and numbers, this is not followed consistently (Line 450 and 453). Moreover, the ml should be written scientifically correct as mL (SI unit).

Spacing and the use of IS standard was checked and changed.

2. Figure 7: is not meaningful, can be put in supplementary files or deleted.

Figure 7 is deleted.

3. Table 8: At the bottom two colored designs are included, but they have no meaning.

The symbols are explained in the text after Table 8.

4. Line 38: world should not be capitalized.

Corrected

5. Line 51: Sentence is grammatically wrong. It should be “authorities authorize use of pesticides only after the critical evaluation of their”

The ‘use of’’ words is inserted in the grammatically correct sentence to make its meaning clearer.

6. There several cases of sentence lacking coherency and structure. Grammar needs urgent attention.

The text was double-checked and modified to improve clarity.

In addition, to make sure that the English language is correctly used in our article, I requested the help of a native English gentleman, Mr. Roger Wood, who is an internationally recognized expert in methods of analyses and sampling and who chaired the analytical working group of Codex Committee on Methods of Analyses and Sampling for over 20 years. Mr. Wood declared the paper well written and suggested only some minor corrections which were naturally accepted.

Reviewer 2 Report

The author answered the question and can be accepted in the present form.

Author Response

Dear Reviewer,

Thank you very much for your positive evaluation.